# Ubiquitous quantum scarring does not prevent ergodicity

Saúl Pilatowsky-Cameo [1], David Villaseñor [1], Miguel A. Bastarrachea-Magnani [2,3], Sergio Lerma-Hernández [4], Lea F. Santos [5✉] & Jorge G. Hirsch [1✉]

In a classically chaotic system that is ergodic, any trajectory will be arbitrarily close to any point of the available phase space after a long time, filling it uniformly. Using Born's rules to connect quantum states with probabilities, one might then expect that all quantum states in the chaotic regime should be uniformly distributed in phase space. This simplified picture was shaken by the discovery of quantum scarring, where some eigenstates are concentrated along unstable periodic orbits. Despite that, it is widely accepted that most eigenstates of chaotic models are indeed ergodic. Our results show instead that all eigenstates of the chaotic Dicke model are actually scarred. They also show that even the most random states of this interacting atom-photon system never occupy more than half of the available phase space. Quantum ergodicity is achievable only as an ensemble property, after temporal averages are performed.

[1] Instituto de Ciencias Nucleares, Universidad Nacional Autónoma de México, Mexico City, Mexico. [2] Department of Physics and Astronomy, Aarhus University, Ny Munkegade, Aarhus C, Denmark. [3] Departamento de Física, Universidad Autónoma Metropolitana-Iztapalapa, Mexico City, Mexico. [4] Facultad de Física, Universidad Veracruzana, Circuito Aguirre Beltrán s/n, Xalapa, Veracruz, Mexico. [5] Department of Physics, Yeshiva University, New York, NY, USA. ✉email: lsantos2@yu.edu; hirsch@nucleares.unam.mx

A striking feature of the quantum-classical correspondence not recognized in the early days of the quantum theory is the repercussion that measure-zero structures of the classical phase space may have in the quantum domain. A recent example is the effect of unstable fixed points, that cause the exponentially fast scrambling of quantum information in both integrable and chaotic quantum systems[1–5]. Another better known example is the phenomenon of quantum scarring[6–8]. As a parameter of a classical system is varied and it transits from a regular to a chaotic regime, periodic orbits that may be present in the phase space change from stable to unstable. These classical unstable periodic orbits can get imprinted in the quantum states as regions of concentrated large amplitudes known as quantum scars. Even though the phase space may be densely filled with unstable periodic orbits, they are still of measure zero, which explains why it took until the works by Gutzwiller[9] for their importance in the quantum chaotic dynamics to be finally recognized.

Quantum scarring was initially observed in the Bunimovich stadium billiard[10] and soon in various other one-body systems[11–13] giving rise to a new line of research in the field of quantum chaos[8,14–22]. The recent experimental observation of long-lived oscillations in chains of Rydberg atoms[23], associated with what is now called "many-body quantum scars", has caused a new wave of fascination with the phenomenon of quantum scarring[24–28]. While the interest in many-body quantum scars lies in their potential as resources to manipulate and store quantum information, a direct relationship between them and possible structures in the classical phase space has not yet been established.

Halfway between one-body and many-body models, one finds systems such as two-dimensional harmonic oscillators and the Dicke model[29], where quantum scars have also been observed[30,31]. In the first case, the model is not fully chaotic and scarring can be understood as an extension of the regular orbits[32,33]. The Dicke model, on the other hand, has a region of strong chaos, where the Lyapunov exponents are positive and the level statistics agrees with the predictions from random matrix theory[34]. The model describes a large number of two-level atoms that interact collectively with a quantized radiation field and was first introduced to explain the phenomenon of superradiance[35,36]. It has been studied experimentally with cavity assisted Raman transitions[37,38], trapped ions[39,40], and superconducting circuits[41].

In this work, we investigate the intricate relationship between quantum scarring and phase-space localization in the superradiant phase of the Dicke model. Even though both phenomena are often treated on an equal footing, the connection is rather subtle. Scarring refers to structures that resemble periodic orbits in the phase-space distribution of quantum eigenstates, while phase-space localization implies that a state exhibits a low degree of spreading in the phase space. Here, we demonstrate that scarring does not necessarily imply significant phase-space localization.

In systems studied before, scarred eigenstates were thought to be a fraction of the total number of eigenstates. In contrast to that, we show that deep in the chaotic regime of the Dicke model, all eigenstates are scarred. Their phase-space probability distributions always display structures that can be traced back to periodic orbits in the classical limit. Yet, we find eigenstates that are highly localized in phase space and eigenstates that are nearly as much spread out as random states, although none of them, including the random states, can cover more than approximately half of the available phase space.

In addition to the analysis of quantum scarring and phase-space localization, we also provide a definition of quantum ergodicity. This is done using a measure that we introduce to quantify the level of localization of quantum states in the phase

space. We say that a quantum state is ergodic if its infinite-time average leads to full delocalization. Under this definition, stationary quantum states are never ergodic, while random states are, and coherent states lie somewhere in between.

## Results

**Dicke model and chaos.** The Hamiltonian of the Dicke model is written as

$$\hat{H}_D = \omega \hat{a}^\dagger \hat{a} + \omega_0 \hat{J}_z + \frac{2\gamma}{\sqrt{\mathcal{N}}} \hat{J}_x(\hat{a}^\dagger + \hat{a}), \qquad (1)$$

where $\hbar = 1$. It describes $\mathcal{N}$ two-level atoms with atomic transition frequency $\omega_0$ interacting with a single mode of the electromagnetic field with radiation frequency $\omega$. In the equation above, $\hat{a}$ ($\hat{a}^\dagger$) is the bosonic annihilation (creation) operator of the field mode, $\hat{J}_{x,y,z} = \frac{1}{2}\sum_{k=1}^{\mathcal{N}} \hat{\sigma}_{x,y,z}^k$ are the collective pseudo-spin operators, with $\hat{\sigma}_{x,y,z}$ being the Pauli matrices, and $\gamma$ is the atom-field coupling strength.

The eigenvalues $j(j+1)$ of the squared total spin operator $\hat{\mathbf{J}}^2 = \hat{J}_x^2 + \hat{J}_y^2 + \hat{J}_z^2$ specify the different invariant subspaces of the model. We use the symmetric atomic subspace defined by the maximum pseudo-spin $j = \mathcal{N}/2$, which includes the ground state. When the Dicke model reaches the critical value $\gamma_c = \sqrt{\omega\omega_0}/2$, it goes from a normal phase ($\gamma < \gamma_c$) to a superradiant phase ($\gamma > \gamma_c$). Our studies are done in the superradiant phase, $\gamma = 2\gamma_c$, and we choose $\omega = \omega_0 = 1$. The rescaled energies are denoted by $\epsilon = E/j$. For the selected parameters, $\epsilon_{GS} = -2.125$ is the ground-state energy.

The classical Hamiltonian, $h_{cl}(\mathbf{x})$ in the coordinates $\mathbf{x} = (q, p; Q, P)$, is obtained by calculating the expectation value of the quantum Hamiltonian under the product of bosonic Glauber and pseudo-spin Bloch coherent states $|\mathbf{x}\rangle = |q, p\rangle \otimes |Q, P\rangle$ (see Methods) and dividing it by $j$. The effective Planck constant $\hbar_{eff} = 1/j$[42] determines the resolution of the quantum states on the four dimensional phase space $\mathcal{M}$. We are able to work with large system sizes ($j \sim 100$ and Hilbert space dimensions $D \sim 6 \times 10^4$), due to the use of an efficient basis that guarantees the convergence of a broad range of eigenvalues and eigenstates[43,44] (see Methods). The Dicke model displays regular and chaotic behavior. For the Hamiltonian parameters selected in this work, the system is in the strong-coupling hard-chaos regime for $\epsilon > -0.8$ (see Supplementary Note 1).

**Quantum scarring.** The (unnormalized) Husimi function of a state $\hat{\rho}$ is defined as $\mathcal{Q}_{\hat{\rho}}(\mathbf{x}) = \langle \mathbf{x}|\hat{\rho}|\mathbf{x}\rangle$, which is the expectation value of the density matrix over the coherent state $|\mathbf{x}\rangle$ centered at $\mathbf{x}$. This function is used to visualize how the state $\hat{\rho}$ is distributed in the phase space. Quantum scars are localized around the classical periodic orbits in an energy shell of the phase space. To visualize the scars, we consider the Husimi projection over the classical energy shell at $\epsilon$,

$$\widetilde{\mathcal{Q}}_{\epsilon,\hat{\rho}}(Q, P) = \iint dq\,dp\, \delta(\epsilon - h_{cl}(\mathbf{x})) \mathcal{Q}_{\hat{\rho}}(\mathbf{x}). \qquad (2)$$

By integrating out the bosonic variables $(q, p)$, the remaining function can be compared with the projection of the classical periodic orbits on the plane of atomic variables $(Q, P)$.

Identifying all periodic orbits that generate the scars of a quantum system is extremely challenging. We were able to identify two families of periodic orbits for the Dicke model, which we denote by family $\mathcal{A}$ and family $\mathcal{B}$[45]. By calculating the overlap of the eigenstates with tubular phase-space distributions located around these orbits[8], we selected twelve eigenstates $\hat{\rho}_k = |E_k\rangle\langle E_k|$ scarred by those two different families. In Fig. 1 we plot their

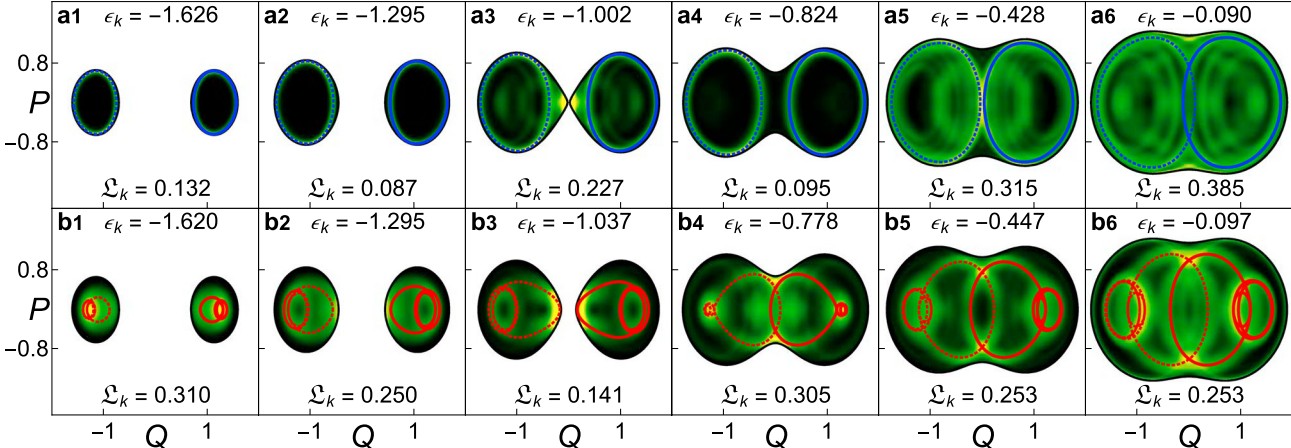

**Fig. 1 Classical periodic orbits and scars in the Husimi projection of eigenstates. a1–a6, b1–b6** Projected Husimi distribution $\widetilde{\mathcal{Q}}_k(Q,P)$ superposed by periodic orbits from the family $\mathcal{A}$ (blue lines) [family $\mathcal{B}$ (red lines)]. Dashed lines mark the mirror image (in $Q$ and $q$) of the periodic orbits drawn with solid lines. The mirror images are also periodic orbits due to the parity symmetry of the Hamiltonian. In the projected Husimi distributions, lighter colors indicate higher concentrations, while black corresponds to zero. The values of the energy $\epsilon_k$ and localization measure $\mathfrak{L}_k$ of each eigenstate $k$ are indicated in the panels. Energies larger than $-0.8$ are in the chaotic region. The system size is $j = 30$.

Husimi projections $\widetilde{\mathcal{Q}}_k = \widetilde{\mathcal{Q}}_{\epsilon_k, \hat{\rho}_k}$ at $\epsilon_k = E_k/j$ along with the corresponding periodic orbit of each family. Family $\mathcal{A}$ (solid blue line in Fig. 1a1–a6) contains the periodic orbits of lowest period of the Dicke model, which emanate from one of the two normal modes around a stable stationary point at the ground-state energy. Family $\mathcal{B}$ (solid red line in Fig. 1b1–b6) arises from the other normal mode around the same point. Scarring is clearly visible in all panels of Fig. 1. The quantum states are highly concentrated around the classical periodic orbits. This happens even in the chaotic region of high excitation energy, where the classical dynamics is ergodic, as seen in Fig. 1a5, a6, b5, b6. Notice that the eigenstates may be scarred by more than one periodic orbit. In fact, as we showed quantitatively in ref. [45] after introducing a measure of scarring, an eigenstate may even be scarred by periodic orbits of different families. This means that different eigenstates exhibit different degrees of scarring.

It is evident from Fig. 1 that the degree of delocalization of the eigenstates in phase space also varies. The Husimi distribution of the eigenstates in Fig. 1a5, a6, for instance, is not entirely confined to the two periodic orbits drawn in blue. This contrasts with the high density concentration that the eigenstate in Fig. 1a4 shows around the plotted unstable periodic orbits. To quantify these differences, we introduce a measure of the degree of localization of a quantum state in the classical energy shell.

**Scarring vs. phase-space localization.** To measure the localization of a state in a Hilbert-space basis indexed by some letter $n$, the most commonly used quantity is the participation ratio $P_R$ [46–48]. Its inverse is given by $P_R^{-1} = \sum_n \mathcal{P}_n^2$, where $\mathcal{P}_n$ is the probability of finding the state in the $n$'th basis vector. By analogy, we introduce a measure of localization in phase space that employs as basis the overcomplete set of coherent states within a single energy shell $\mathcal{M}_\epsilon = \{\mathbf{x} = (q, p; Q, P) | h_{cl}(\mathbf{x}) = \epsilon\}$, so that we replace the sum $\Sigma_k$ with a three-dimensional surface integral $\int_{\mathcal{M}_\epsilon} d\mathbf{s}$ over $\mathcal{M}_\epsilon$. For a given $\mathbf{x} \in \mathcal{M}_\epsilon$, the probability $\mathcal{P}_\mathbf{x}$ of finding the state $\hat{\rho}$ in the coherent state $|\mathbf{x}\rangle$ is given by the Husimi function $\mathcal{Q}_{\hat{\rho}}(\mathbf{x})$. With these replacements, we finally obtain a measure of phase-space localization $\mathfrak{L}(\epsilon, \hat{\rho})$ given by

$$\mathfrak{L}(\epsilon, \hat{\rho})^{-1} = \frac{1}{N} \int_{\mathcal{M}_\epsilon} d\mathbf{s} \; \mathcal{Q}_{\hat{\rho}}^2(\mathbf{x}), \qquad (3)$$

where $N = (\int_{\mathcal{M}_\epsilon} d\mathbf{s} \; \mathcal{Q}_{\hat{\rho}}(\mathbf{x}))^2 / \mathcal{V}(\epsilon)$ is a normalization constant and $\mathcal{V}(\epsilon) = \int_{\mathcal{M}_\epsilon} d\mathbf{s}$ is the volume of $\mathcal{M}_\epsilon$ (see Methods).

The measure $\mathfrak{L}(\epsilon, \hat{\rho})$ is an energy-restricted second moment of the Husimi function [49]. It is related to the second-order Rényi-Wehrl entropy [47], which, in turn, was shown for the Dicke model [50] to be linearly related to the first-order Rényi-Wehrl entropy [51].

The value of $\mathfrak{L}(\epsilon, \hat{\rho})$ indicates the fraction of the classical energy shell at $\epsilon$ that is covered by the state $\hat{\rho}$. It varies from its minimum value $\mathfrak{L}(\epsilon, \hat{\rho}) \sim (2\pi\hbar_{eff})^2 / \mathcal{V}(\epsilon)$, which indicates maximum localization, to $\mathfrak{L}(\epsilon, \hat{\rho}) = 1$, which implies complete delocalization over the energy shell. The former occurs for coherent states, and the latter happens if $\mathcal{Q}_{\hat{\rho}}(\mathbf{x})$ is a constant for all $\mathbf{x} \in \mathcal{M}_\epsilon$, in which case the projection $\widetilde{\mathcal{Q}}_{\epsilon, \hat{\rho}}(Q, P)$ is also constant for the allowed values of $Q$ and $P$.

All eigenstates in Fig. 1 have values of $\mathfrak{L}_k = \mathfrak{L}(\epsilon_k, \hat{\rho}_k)$ below $1/2$. For the eigenstates in Fig. 1a1, a2, a4, b3, these values are very small, since the eigenstates are almost entirely localized around the plotted periodic orbits. The value of $\mathfrak{L}_k$ is larger in Fig. 1a3, because at the center of the diagram, there is an unstable stationary point [2], which produces a one-point scar in addition to the scar associated to the blue orbit. The localization measure is larger in Fig. 1b1, b2 simply because in these cases the phase space is very small. It is also larger for the states in the high energy region in Fig. 1a5, a6, b4, b5, b6, because they spread beyond the marked periodic orbits. As the energy increases and one approaches the chaotic region, more unstable periodic orbits emerge in the classical limit, enhancing the likelihood that a single quantum eigenstate gets scarred by different periodic orbits. We stress, however, that even for those high-energy states with larger values of $\mathfrak{L}_k$, the drawn periodic orbits cast a bright green shadow that is clearly visible in the Husimi projections.

It is important to make it clear that scarring and localization, despite related, are not synonyms. Of course, there is no eigenstate with a large value of $\mathfrak{L}_k$ that would at the same time have a high value of our scarring measure, but there is more to the relationship between these two concepts. A highly localized eigenstate is scarred by few periodic orbits of a particular family and therefore has a high value of the scarring measure for that family, although it has low values for the other families of periodic orbits [45]. Furthermore, even the most delocalized

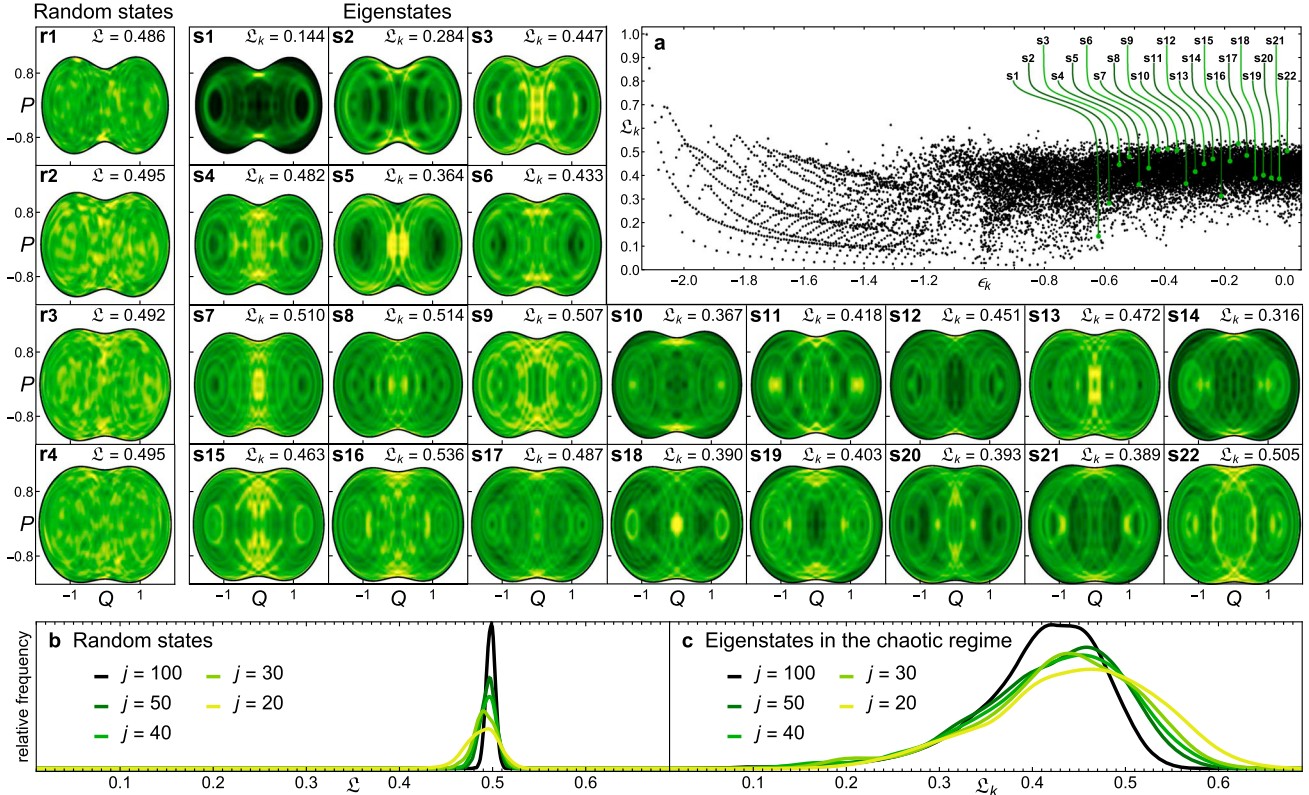

**Fig. 2 Husimi projection and localization measure of eigenstates and random states. a** Localization measure $\mathfrak{L}_k$ as a function of energy for the first 17,000 eigenstates with $\epsilon_k \in [\epsilon_{GS}, 0.1]$, $j = 100$. **(s1-s22)** Projected Husimi distributions for 22 eigenstates selected every 400 values of $k$, starting at $k = 7700$ in (s1) up to $k = 16500$ in (s22), $j = 100$. Lighter colors indicate higher concentrations, while black corresponds to zero. **(r1-r4)** Projected Husimi distributions for random states of energy width $\sigma = 0.3$ centered at $\epsilon = -0.6$ in (r1), $\epsilon = -0.4$ in (r2), $\epsilon = -0.2$ in (r3), and $\epsilon = -0.1$ in (r4), $j = 100$. Lighter colors indicate higher concentrations, while black corresponds to zero. **b** Distribution of $\mathfrak{L}$ for 20,000 random states of width $\sigma = 0.3$ centered at $\epsilon = -0.5$. **c** Distribution of $\mathfrak{L}_k$ for the eigenstates in the chaotic region with $\epsilon_k \in [-0.8, 0]$. The system sizes for **b** and **c** are indicated in the panels.

eigenstates are still scarred, but now by periodic orbits from different families[22].

In Fig. 2, we take a step further in the analysis of localization and scarring. In the large panel in Fig. 2a, we show $\mathfrak{L}_k$ against energy for all eigenstates between $\epsilon_{GS}$ and $\epsilon = 0.06$. This plot is equivalent to a Peres lattice[52] for expectation values of observables, as used in studies of chaos and thermalization. In the low-energy regular regime, $\mathfrak{L}_k$ is organized along lines that can be classified with quasi-integrals of motion linked with classical periodic orbits[53]. Conversely, as the system enters the chaotic region at higher energies, the distribution of $\mathfrak{L}_k$ becomes dense and looses any order. Notice, however, that all eigenstates in the chaotic region have values of $\mathfrak{L}_k$ much lower than 1, mostly clustering below 1/2.

The value $\mathfrak{L} \sim 1/2$ marks a limit on the spreading of any pure state in the high energy shells of the phase space. To show this, we build random states $|\mathcal{R}_\epsilon\rangle = \sum_k c_k |E_k\rangle$, where $c_k$ are complex random numbers from a Gaussian energy profile centered at energy $\epsilon$ (see Methods). The values of the localization measure $\mathfrak{L}(\epsilon, \hat{\rho}_\epsilon)$ for four different random states $\hat{\rho}_\epsilon = |\mathcal{R}_\epsilon\rangle\langle\mathcal{R}_\epsilon|$ centered at increasing energies $\epsilon$ between $-0.6$ and $-0.1$ are given in Fig. 2r1-r4, and indeed $\mathfrak{L} \sim 1/2$. This upper bound on the phase-space delocalization is not due to quantum scarring, but to quantum interference effects.

The panels in Fig. 2r1–r4 display the projected Husimi distributions $\widetilde{\mathcal{Q}}_{\epsilon, \hat{\rho}_\epsilon}(Q, P)$ of the four random states and those in Fig. 2s1-s22 show the distributions for 22 eigenstates taken at fixed steps of $k$ with $\epsilon_k \in [-0.62, 0.06]$ (various other examples are provided in the Supplementary Note 2). The difference

between random states and eigenstates is clear. The Husimi projections of the random states do not show structures that resemble closed periodic orbits, so they are not scarred, while the Husimi projections of all eigenstates in the chaotic region do show those structures. In contrast to Fig. 1 (see also ref. [45]), we have not identified the periodic orbits associated with Fig. 2s1-s22, but the visible circular patterns are clear evidence of periodic orbits. Their existence is supported by the shape of the Husimi projections and by knowing the generic direction of the classical Hamiltonian flow. The patterns display all the features of periodic orbits: they always cross the line $P = 0$ perpendicularly, they are symmetric along both the horizontal $Q$ and $P$ axes, and they visibly form closed loops. There is no quantum effect other than scarring that would produce such patterns. One therefore deduces that deep in the chaotic region of the Dicke model, all quantum eigenstates are scarred, although they have different degrees of scarring, as seen by the patterns, and they have different degrees of localization, ranging from strong localization ($\mathfrak{L} \sim 0.1$) to states that are nearly as much delocalized as random states.

Our results are sharpened as one approaches the semiclassical limit. The patterns indicating scarring do not fade away as the system size increases. Quite the opposite, as $j = 1/\hbar_{\text{eff}}$ increases, the periodic orbits get better defined in the Husimi projections (cf. the figures for $j = 30$ and $j = 100$ in the Supplementary Note 3). To study the dependence of the level of phase-space localization on system size, we show the distributions of $\mathfrak{L}$ for random states (Fig. 2b) and for eigenstates in the chaotic region (Fig. 2c) for different values of $j$. For the random states, $\mathfrak{L}$ is concentrated around 1/2 and the width of the distribution

decreases as $j$ increases, corroborating that $\mathfrak{L} = 1/2$ is indeed the delocalization upper bound. In contrast, for the eigenstates in the chaotic regime, the distributions are skewed and broader. The tail at small values of $\mathfrak{L}_k$ does not change as $j$ increases, showing that the highly localized states persist, while the portion of the states with large $\mathfrak{L}_k$ decreases, suggesting that for large system sizes, none of the eigenstates reach values of $\mathfrak{L}_k > 1/2$.

The ubiquitous scarring revealed by our studies motivates the question of whether scarring in other quantum models is also the rule. We have found hints in the literature suggesting that our findings may actually be quite general. For example, in ref. [22], the authors reconstruct significant portions of the spectrum of a quantum chaotic system using only periodic orbits. This means that all of these eigenstates are described by those periodic orbits and are therefore scarred. In ref. [20], the authors claim that the great majority of the eigenstates of the hydrogen atom in a magnetic field may be related to periodic orbits, indicating that scars must be the rule. But to provide a definite answer, a phase-space analysis similar to the one presented here is needed.

**Quantum ergodicity.** We have so far discussed two concepts – quantum scarring and phase-space localization – that are related, but are not equal. How about their relationship with quantum ergodicity? In the classical limit, a system is ergodic if the trajectories cover the energy shell homogeneously. We then adopt the same definition for quantum ergodicity. To quantify how much of the energy shell is visited on average by the evolved state $\hat{\rho}(t) = e^{-i\hat{H}_D t}\hat{\rho}\, e^{i\hat{H}_D t}$, we consider the infinite-time average[54,55],

$$\overline{\rho} = \lim_{T\to\infty} \frac{1}{T}\int_0^T \mathrm{d}t\; \hat{\rho}(t), \tag{4}$$

and compute $\overline{\mathfrak{L}}(\epsilon,\hat{\rho}) \equiv \mathfrak{L}(\epsilon,\overline{\rho})$ with Eq. (3). If the whole energy shell at $\epsilon$ is homogeneously visited by $\hat{\rho}$, then $\overline{\mathfrak{L}}(\epsilon,\hat{\rho}) = 1$. We thus say that a quantum state $\hat{\rho}$ is ergodic over the energy shell $\epsilon$ if $\overline{\mathfrak{L}}(\epsilon,\hat{\rho}) = 1$. According to this definition, all stationary states in the chaotic region of the Dicke model are non-ergodic, since $\overline{\mathfrak{L}}(\epsilon_k,\hat{\rho}_k) = \mathfrak{L}(\epsilon_k,\hat{\rho}_k) \lesssim 1/2$, as shown above.

How about non-stationary states, such as coherent states or random states, are they ergodic? We study the evolution of initial coherent states $|\Psi(0)\rangle = |\mathbf{x}_0\rangle = \sum_k c_k |E_k\rangle$ with mean energies $\epsilon = -0.5$, that are in the chaotic region (see Methods). We select both coherent states that are highly localized and delocalized in the energy eigenbasis, with the degree of delocalization measured by the participation ratio $P_R = \left(\sum_k |c_k|^4\right)^{-1}$. Their energy distributions are shown in Fig. 3a1–g1. The components of the states with low $P_R$ are bunched around specific energy levels (Fig. 3a1, b1), exhibiting the comb-like structure typical of scarred states[6]. As $P_R$ increases, the coherent states become more spread in the energy eigenbasis, looking more similar to the random state $|\Psi(0)\rangle = |\mathcal{R}_\epsilon\rangle$ shown in Fig. 3h1, whose mean energy is also $\epsilon = -0.5$.

The evolution of the survival probability, $S_P(t) = |\langle\Psi(0)|e^{-i\hat{H}_D t}|\Psi(0)\rangle|^2$, for the coherent states with low $P_R$ leads to large revivals before the saturation of the dynamics[56], as seen in Fig. 3a2, b2, c2. This contrasts with the evolution of the coherent states with large $P_R$, such as those in Fig. 3d2–g2, and the evolution of the random state in Fig. 3h2. In these cases, the approach to the asymptotic value of $S_P(t)$ is much smoother and exhibits the so-called correlation hole, which corresponds to the ramp towards saturation. The correlation hole reflects the presence of correlated eigenvalues and is a quantum signature of classical chaos[57–59].

The values of $\overline{\mathfrak{L}}(\epsilon,\hat{\rho})$ for the states in Fig. 3a1–h1 are indicated in Fig. 3a3–h3. The random state is indeed ergodic, reaching $\overline{\mathfrak{L}} \sim 1$. For the coherent states, $\overline{\mathfrak{L}}$ increases as $P_R$ does, but even

for the states with the largest values of the participation ratio, $\overline{\mathfrak{L}}$ is still slightly under 1. The analysis of the dependence of $\overline{\mathfrak{L}}$ on system size is done in Fig. 3i, j. The distribution of $\overline{\mathfrak{L}}$ for random states (Fig. 3j) gets narrower and better centered at $\overline{\mathfrak{L}} = 1$ as $j$ increases, confirming that these states indeed behave ergodically. The distribution for the coherent states (Fig. 3h) is much broader. The center moves towards larger values as $j$ increases, but it is not clear whether there will ever be a significant portion of the initial coherent states with $\overline{\mathfrak{L}} \sim 1$.

In Fig. 3a3–h3, we plot the projected time-averaged Husimi distributions $\widetilde{\mathcal{Q}}_{\epsilon,\overline{\rho}}(Q,P)$ for the states in Fig. 3a1–h1. Remarkably, even for the coherent states with high $P_R$, which do not exhibit any comb-like structure in their energy distributions and do not show revivals in the evolution of their survival probability, we still see an enhancement around unstable periodic orbits, as revealed by a careful inspection of Fig. 3d3–g3 and in contrast with the absence of any pattern for the random state in Fig. 3h3. This unexpected manifestation of dynamical scarring can only be observed in phase space, having no identifiable signature in the energy distribution of the initial states or in the evolution of the survival probability. Thus, revivals in the long-time quantum dynamics are signs of a scarred initial state, but lack of revivals do not exclude the presence of scarring, it just indicates that if it exists it is at a low degree.

## Discussion

The three main concepts investigated and compared in this work were quantum scarring, phase-space localization, and quantum ergodicity. We showed that for the Dicke model, all eigenstates in the chaotic region are scarred, although with different degrees of scarring and different levels of phase-space localization. Evidently, an eigenstate that is strongly scarred is also highly localized in phase space, but a single eigenstate may be scarred by different periodic orbits and reach levels of delocalization almost as high as a random state. We also showed that any pure state – even without any trace of scarring – is localized in phase space, and that ergodicity is an ensemble property, achievable only through temporal averages. Thus, scarring, localization and lack of ergodicity are not synonyms, although connections exist.

The ubiquitous scarring of the eigenstates does not immediately translate into the breaking of quantum ergodicity. All eigenstates are certainly non-ergodic, since they never reach complete delocalization in phase space, but if a non-stationary state visits on average the available phase-space homogeneously, then it is ergodic. Random states, for example, are ergodic. The analysis of initial coherent states showed that some are heavily scarred, resulting in the strong breaking of ergodicity that translates into the usual revivals of the survival probability. More interesting is the subtle behavior of the majority of the initial coherent states, which do not display revivals in the quantum dynamics or the comb-like structure in their energy distributions, but yet show some degree of scarring.

Analyses that focus on the Hilbert space, such as the energy distribution of the initial states or the fluctuations of eigenstate expectation values in Peres lattices and comparisons with thermodynamic averages, that are often done in studies of the eigenstate thermalization hypothesis (ETH), may miss the ubiquitous scarring observed in this work. For this feature to be revealed, one needs to look at the structures of the states in phase space.

Our results for the Dicke model are, of course, important, due to the widespread theoretical interest in this model and the fact that it is employed to describe experiments with trapped ions and

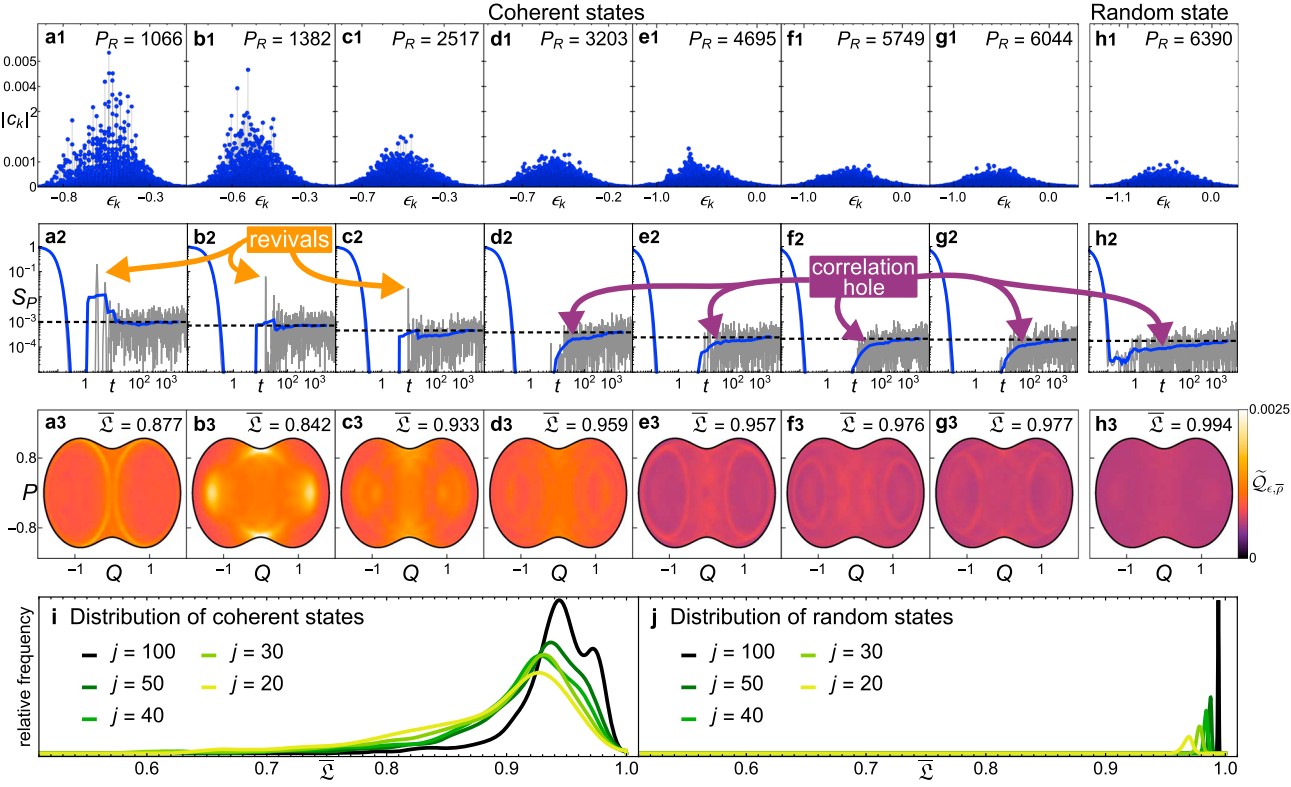

**Fig. 3 Dynamical behavior of coherent and random states. a1–g1** Energy distribution for coherent states with different values of $P_R$ and centered at $\epsilon = -0.5$, $j = 100$. **h1** Energy distribution for a random state $\left|\mathcal{R}_\epsilon\right\rangle$ with energy width $\sigma = 0.3$ centered at $\epsilon = -0.5$, $j = 100$. **a2–h2** Quantum survival probability (gray solid line), its running average (blue solid line) and its asymptotic value (black dashed line) for the corresponding states of panels **a1–h1**. **a3–h3** Projected Husimi distributions of the time-averaged ensemble, $\widetilde{\mathcal{Q}}_{\epsilon=-0.5,\overline{p}}$ for the corresponding states of panels a1–h1 with the values of $\overline{\mathfrak{L}}(\epsilon = -0.5, \hat{\rho})$ indicated. **i** Distribution of $\overline{\mathfrak{L}}(\epsilon, \hat{\rho})$ for a set of 1551 coherent states that are evenly distributed along the atomic variables $(Q, P)$ at $\epsilon = -0.5$ and $p = 0$. This is done for system sizes $j = 20, 30, 40, 50, 100$, as indicated. **j** Distribution of $\overline{\mathfrak{L}}(\epsilon, \hat{\rho})$ for 500 random states with $\epsilon = -0.5$ and $\sigma = 0.3$ for $j = 20, 30, 40, 50, 100$.

superconducting circuits. But the repercussion of the findings discussed here goes beyond the limit of spin-boson systems by raising the question of whether scarring in other quantum systems is also the rule and not the exception. Our work provides the appropriate tools to address this question. The phase-space method that we developed is applicable to any quantum system that has a tractable phase-space. Whether these studies could eventually be extended also to interacting many-body quantum systems, such as interacting spin-1/2 models, will depend on the viability of their semiclassical analysis, and some recent works give reasons for optimism[60–62].

## Methods

**Classical Hamiltonian.** To construct the classical Hamiltonian in a four-dimensional phase space $\mathcal{M}$ with coordinates $\mathbf{x} = (q, p; Q, P)$, we use the Glauber-Bloch coherent states $|\mathbf{x}\rangle = |q, p\rangle \otimes |Q, P\rangle$[30,34,63–66]. They are tensor products of the bosonic Glauber coherent states $|q, p\rangle = e^{-(j/4)(q^2+p^2)} e^{\left[\sqrt{j/2}(q+ip)\right]\hat{a}^\dagger} |0\rangle$ and the pseudo-spin Bloch coherent states $|Q, P\rangle = \left(1 - \frac{Z^2}{4}\right)^j e^{[(Q+iP)/\sqrt{4-Z^2}]\hat{J}_+} |j, -j\rangle$, where $Z^2 = Q^2 + P^2$, $|0\rangle$ denotes the photon vacuum, $|j, -j\rangle$ designates the state with all atoms in their ground state, and $\hat{J}_+$ is the raising atomic operator. The classical Hamiltonian is given by

$$h_{cl}(\mathbf{x}) = \frac{\omega}{2}(q^2 + p^2) + \frac{\omega_0}{2} Z^2 + 2\gamma Qq\sqrt{1 - \frac{Z^2}{4}} - \omega_0. \tag{5}$$

**Efficient basis and system sizes.** The efficient basis is the Dicke Hamiltonian (1) eigenbasis in the limit $\omega_0 \to 0$, which can be analytically obtained by a displacement of the bosonic operator $\hat{A} = \hat{a} + (2\gamma/(\omega\sqrt{\mathcal{N}}))\hat{J}_x$ and a rotation of $-\pi/2$ around the

$y$ axis of the collective pseudo-spin operators $\left(\hat{J}_x, \hat{J}_y, \hat{J}_z\right) \to \left(\hat{J}'_z, \hat{J}'_y, -\hat{J}'_x\right)$,

$$|N; j, m'\rangle = \frac{\left(\hat{A}^\dagger\right)^N}{\sqrt{N!}} |N = 0; j, m'\rangle, \tag{6}$$

where $N$ is the eigenvalue of the modified bosonic number operator $\hat{A}^\dagger\hat{A}$ and $m' = m_x$ is the eigenvalue of the original collective pseudo-spin operator $\hat{J}_x$. The modified bosonic vacuum states in $|N = 0; j, m'\rangle = |N = 0\rangle \otimes |j, m'\rangle$ are Glauber coherent states $|N = 0\rangle = \left|-2\gamma m'/(\omega\sqrt{\mathcal{N}})\right\rangle$. The Hilbert space dimension of this basis is given by $D = (2j + 1)(N_{max} + 1)$, where $N_{max}$ designates a cutoff of the modified bosonic subspace.

This basis allows to work with larger values of $j$ by reducing the value of $N_{max}$ required for convergence of the high-energy eigenstates. With $j = 100$ and $N_{max} = 300$ ($D = 60,501$), we are able to get 30,825 converged eigenstates covering the whole energy spectrum up to $\epsilon = 0.853$. Having converged eigenstates in such a high-energy regime would be infeasible with the usual Fock basis for $j = 100$[44,67–69].

**Husimi projection and localization measure.** To compute the Husimi projection given in Eq. (2) and the localization measure given by Eq. (3), one has to compute integrals of the form

$$\widetilde{f}(Q, P) = \iint dq\, dp\; \delta(\epsilon - h_{cl}(\mathbf{x})) f(\mathbf{x}), \tag{7}$$

where $\mathbf{x} = (q, p; Q, P)$ and $f(\mathbf{x})$ is a non-negative function in the phase space. For the localization measure, note that $\mathfrak{L}(\epsilon, \hat{\rho})^{-1} = \frac{1}{N} \iint dQ\,dP \widetilde{f}_2(Q, P)$ with $f_2 = \mathcal{Q}_{\hat{\rho}}^2$, $N = \left(\iint dQ\,dP \widetilde{f}_1(Q, P)\right)^2 / \mathcal{V}(\epsilon)$ with $f_1 = \mathcal{Q}_{\hat{\rho}}$, and $\mathcal{V}(\epsilon) = \iint dQ\,dP \widetilde{f}_0(Q, P)$ with $f_0(\mathbf{x}) = 1$.

By using properties of the $\delta$ function, those integrals are reduced to

$$\widetilde{f}(Q,P) = \int_{p_-}^{p_+} dp \frac{\sum_{q_\pm} f(q_\pm, p; Q, P)}{\sqrt{\Delta(\epsilon, p, Q, P)}}, \tag{8}$$

where $q_\pm$ are the two solutions in $q$ of the second-degree equation $h_{cl}(q, p; Q, P) = \epsilon$,

$$\Delta(\epsilon, p, Q, P) = \left| \frac{\partial h_{cl}}{\partial q}(q_\pm, p; Q, P) \right|^2$$
$$= 2\omega\omega_0 \left( \frac{\epsilon}{\omega_0} + 1 - \frac{Q^2 + P^2}{2} \right) + 4\gamma^2 Q^2 \left( 1 - \frac{Q^2 + P^2}{4} \right) - \omega^2 p^2, \tag{9}$$

and $p_\pm$ are the two solutions in $p$ of the second-degree equation $\Delta(\epsilon, p, Q, P) = 0$. Because of the form of the weight $1/\sqrt{\Delta}$, the integral given by Eq. (8) may be computed efficiently using a Chebyshev-Gauss quadrature method.

It is worth noting that a quantum state with a Wigner distribution that is constant in an energy shell will lead to a Husimi function that needs not to be constant within the same energy shell. This is because the Husimi distribution is the convolution of the Wigner distribution with the Gaussian Wigner distributions of the coherent states, which have different energy widths due to the geometry of the energy shells in the phase space. This rather marginal effect may be seen in Fig. 3h3, where there is a barely visible weak concentration towards the center of the plot causing $\overline{\mathfrak{L}}$ to be slightly under 1. We stress that this effect is not related to quantum scarring. It is just a manifestation of the phase-space geometry in the Husimi distributions.

**Coherent states**. The coherent states $|\mathbf{x}\rangle$ are described by four coordinates $\mathbf{x} = (q, p; Q, P)$. To select coherent states at a given energy $\epsilon$, we solve the second-degree equation $h_{cl}(q, p; Q, P) = \epsilon$ for $q$, which yields two solutions $q_+(\epsilon, p, Q, P) \geq q_-(\epsilon, p, Q, P)$[59]. All of the initial coherent states shown in Fig. 3 have bosonic variables given by $q = q_+(\epsilon = -0.5, p, Q, P)$ and $p = 0$. The coherent states shown in Fig. 3a1-g1 have atomic coordinates given by $(Q, P) = (1.75, 0)$ a1, $(0.5, 0.75)$ b1, $(0.75, 0.5)$ c1, $(1.25, 0.25)$ d1, $(-1.25, 1)$ e1, $(-1.25, 0.75)$ f1, and $(-0.75, 0.5)$ g1. The atomic coordinates of the 1551 coherent states whose distributions of $\overline{\mathfrak{L}}$ are shown in Fig. 3i were selected by constructing a rectangular grid with step $\Delta Q = \Delta P = 0.05$ from $(Q_i, P_i) = (-2, -2)(Q_i, P_i) = (-2, -2)$ to $(Q_f, P_f) = (2, 0)$. Of the 3321 points inside of this grid, 1551 have allowed values of $Q, P$ (i.e. they satisfy $Q^2 + P^2 \leq 4$) and fall inside of the energy shell at $\epsilon = -0.5$ (i.e. there exists $q_+(\epsilon = -0.5, p = 0, Q, P)$). We use these 1551 coherent states to compute the distributions in Fig. 3i.

**Random states**. The state $|\mathcal{R}_\epsilon\rangle = \sum_k c_k |E_k\rangle$ is built by sampling random numbers $r_k > 0$ from an exponential distribution $\lambda e^{-\lambda x}$ and random phases $\theta_k$ from a uniform distribution in $[0, 2\pi)$. We use

$$c_k = \sqrt{\frac{r_k \rho(E_k)}{\nu(E_k) M}} \, e^{i\theta_k} \tag{10}$$

where $\nu(E)$ is the density of states, $\rho(E)$ is a Gaussian profile of width $j\sigma$ centered at energy $j\epsilon$, and $M$ ensures normalization. This way, $|\mathcal{R}_\epsilon\rangle$ has a defined energy center $\epsilon$, where the Husimi projection and the localization measure are calculated[59,70].

## Data availability

All the data that support the plots within this paper and other findings of this study are available from the corresponding authors upon request.

## Code availability

All the computational codes that were used to generate the data presented in this paper are available from the corresponding authors upon request.

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

## Acknowledgements

We thank D. Wisniacki for his valuable comments and acknowledge the support of the Computation Center - ICN, in particular of Enrique Palacios, Luciano Díaz, and Eduardo Murrieta. S.P.-C., D.V. and J.G.H. acknowledge financial support from the DGAPA-UNAM project IN104020, and SL-H from the Mexican CONACyT project CB2015-01/255702. LFS was supported by the NSF grant No. DMR-1936006.

## Author contributions
S.P.-C. and D.V. were responsible for most of the calculations and the development of the work. M.A.B.-M., S.L.-H., L.F.S. and J.G.H. provided the original ideas and shaped the manuscript.

## Competing interests
The authors declare no competing interests.
