## [Peer Review File · Nature Communications]

Reviewers' Comments:

Reviewer #1:

Remarks to the Author:

The manuscript provides an interesting example of excessive scarring of eigenstates by unstable periodic orbits in the regime of parameters corresponding to classically chaotic dynamics. This subject has been extensively studied in the past notably in 80's and 90's of the past century but obtained some revival in a different context of many-body dynamics recently. The authors consider a ``classical'' scenario of two degrees of freedom, a large collective spin (actually corresponding to large number of two level atoms but this is not relevant for a theory) driven by a strong single harmonic mode without the rotating wave approximation. The system is considered in the so called superradiant regime.

Authors raise an interesting and timely question – whether scarring prevents ergodicity? They define a delocalization measure (quite complicated in fact), say L , which should be unity for states delocalized over the energy shell. However, they find numerically that for the system studied, in the chaotic regime, L saturates at 0.5. First they link it to excessive scarring revealed in nice pictures – but later they show that even random states do not cross this limit. On the other hand the delocalization may reach (Fig.2) 0.7 for states in the regular regime. This fact is later used to discuss ergodicity using nonstationary state dynamics. While initially localized wavepackets reveal traces of revivals in survival probability, the delocalized ones reveal a characteristic correlation hole (well known from eighties again). Since the authors have at their disposal L calculated for a number of states one could naturally ask what is L distribution in the chaotic regime? What could be said about it from some approach based on random matrices? There are other known measures for eigenstates – if indeed states of this system are so characteristic (excessive scarring) – does it show in other statistical properties of eigenstates? How general are these results (see below)?

While exemplary results presented are interesting no definite conclusions are formed, the manuscript brings observations and questions but few definite answers. In this respect it may stimulate discussion. However, the model studied is quite specific and studied by the same group in a number of papers (say 10 or more) in the last 6 years.

Different aspects of spectral and dynamical properties were studied including scarring, Lyapunov exponents etc. In particular spectral correlations and scarring were studied in an almost parallel paper in NJP 22 (2020) 063036 – stressing, however, different aspects than this work.

In the eighties scarring was studied not only for billiards or kicked top – incidentally for the latter the first reference on scars, prior to [13] is M. Kus, J. Zakrzewski and K. Zyczkowski, Phys. Rev. A43, 4244 (1991). More importantly, pioneering works on scarring in hydrogen atom in magnetic field, which really introduced periodic orbit theory to experiments should be cited (in particular works of late D. Wintgen, see e.g. D. Wintgen and A. Hönl Phys. Rev. Lett. 63, 1467 (1989) and later works e.g. K. Müller and D. Wintgen J. Phys. B 91994), works with the experimental group of late Welge). There is also an interesting Physics Reports from 1993 of Bohigas Tomsovic and Ullmo on a related topic, certainly worth a citation.

Looking at Fig.2 or supplementary material one hesitates whether the scarring is not enhanced by the relatively large effective $\hbar=1/j$ in the model studied. Understanding the technical difficulties of the model one wonders whether a study in other models mentioned might not be more conclusive.

Reviewer #2:

Remarks to the Author:

In the manuscript "Does scarring prevent ergodicity?", the authors study the Dicke model's eigenstate properties and dynamics in detail. Using a phase space approach, they find that all of the eigenstates never occupy the full phase-space and, more than that, exhibit patterns of

unstable periodic orbitals in the classical Hamiltonian. As a result, they claim that *all* eigenstates of the Dick model are scarred, while the level statistics of the model still shows Wigner-Dyson distribution. They further demonstrate that the ergodicity is restored by averaging random state's long-time dynamics.

This paper's results are counterintuitive and potentially very interesting since previous found scarred eigenstates only take up measure zero portion of the whole spectrum. However, so far, the authors just demonstrate the results without much explanation. For example, there is no explanation of why all the eigenstates encode the unstable periodic orbitals in the classical limit, which is the most striking result. Furthermore, because of the phase space method that the authors use, it is difficult to assess whether the results are tied to this particular method or are intrinsic about the eigenstate in the Dicke Model. As the authors also admit, it is not obvious how to identify the phenomena observed here in other quantum systems like spin 1/2 models, where the phase space method is not applicable. Therefore, at this stage, I cannot recommend its publication in nature communication before these issues are addressed. Please find more specific questions and comments below.

1) The measure L introduced in eq (3) to quantify the localization in phase space is not enough to claim that all eigenstates are scarred. None of the pure states occupy the whole phase space, including the random states. L of many eigenstates in the high energy shell is comparable with that of random states ~ 0.5 .

2) The time-averaged L introduced below eq (3) is also not a good measure to claim all eigenstates are scarred. Quantum mechanics determines that the eigenstate does not have dynamics and does not explore the phase space, in contrast to a random state. This measure leads to that all eigenstates of all quantum systems are scarred, which is not meaningful.

3) The most interesting and surprising result I find is that all eigenstates encode the pattern of the unstable periodic orbital, different from a random state. However, the authors did not discuss the mechanism behind this or provide a quantitative measure to quantify the eigenstate's closeness to the periodic orbital. Is there a quantitative measure to distinguish the pattern of phase space in Fig 2 (r4) and (s22), which have a similar L ?

4) Can the result found be a finite-size effect? How does the result change as ω_0 increase?

5) In recent studies of quantum many-body scars related to the long-time oscillation found in the Rydberg atom experiments, a Hallmark of quantum scar is that order L eigenstates violate eigenstate thermalization hypothesis (ETH) while the level statistics of the whole spectrum remains Wigner-Dyson like. Do the scarred states identify here, namely all eigenstates, violate ETH? This can be checked by comparing the local observables measured from eigenstates and the Gibbs ensemble. What does the projected Husimi distribution of a Gibbs state look like? ETH violation can be used as a measure independent of the phase space method to strengthen the authors' claim.

6) Is there any experimentally measurable consequence of these findings?

REPLY TO REVIEWER #1

We thank the Reviewer for carefully reading our work and for finding it interesting and timely. We appreciate the historical account that the Reviewer brought to our attention, along with the very pertinent references, which we incorporated to our manuscript. Below, we reply point-by-point each one of the Reviewer's comments and explain the changes that they motivated to our revised version.

Point-by-point reply:

- “They define a delocalization measure (quite complicated in fact), say L , which should be unity for states delocalized over the energy shell.”

Reply. The measure \mathfrak{L} that we introduced is actually not complicated and it is a very natural way to measure delocalization within single energy shells of the phase space. However, the way we had presented it may indeed appear complicated. We now changed the notation and added a discussion before equation (3) to explain that \mathfrak{L} is a natural extension of the well-known quantum participation ratio. We thank the Reviewer for motivating this improvement, which we are sure will be useful to the readers.

- “Since the authors have at their disposal L calculated for a number of states one could naturally ask what is L distribution in the chaotic regime?”

Reply. This is a very good question. To answer it, we added panel (c) to Fig. 2, which contains the distribution of \mathfrak{L}_k for the chaotic eigenstates for different system sizes ($j = 20, 30, 40, 50, 100$). The distribution is skewed to smaller values of \mathfrak{L}_k . Furthermore, the tail at the low values of \mathfrak{L}_k seems to be independent on the system size, while the largest values of \mathfrak{L}_k become smaller as j increases.

- “What could be said about it from some approach based on random matrices?”

Reply. The eigenstates of random matrices are random vectors. In the new panel (b) of Fig. 2, we show the distribution of \mathfrak{L} for an ensemble of 20,000 random states for $j = 20, 30, 40, 50, 100$. The random states concentrate around $\mathfrak{L} = 1/2$ and the width of the distribution decreases as j increases [Fig. 2 (b)]. This contrasts with the chaotic eigenstates, which have a wider distribution [Fig. 2 (c)] and reach $\mathfrak{L}_k \geq 1/2$ for very few cases. In addition, this portion of eigenstates with $\mathfrak{L}_k \geq 1/2$ decreases as j increases. These results indicate that the behavior of the chaotic eigenstates cannot be explained by that of random states.

Another difference between the eigenstates and the random states is that the Husimi distributions of the former always show patterns typical of periodic orbits, even when the eigenstate has \mathfrak{L}_k close to $1/2$, while these patterns are nonexistent for the random states. We have extended these discussions in our manuscript.

- “There are other known measures for eigenstates – if indeed states of this system are so characteristic (excessive scarring) – does it show in other statistical properties of eigenstates?”

Reply. First, let us say that we do not believe that the ubiquitous scarring found in this work is a unique feature of our model, but we expand on this on the reply to the next point.

Excessive scarring may not be completely revealed by traditional statistical measures associated with the Hilbert space, such as participation ratios or Peres lattices. To uncover the ubiquitous scarring observed in our work, one needs to go to phase space, since the definition of scarring is tied to phase space. We need to have measures of localization with respect to the phase space (such as our \mathfrak{L} or Rényi-Wherl entropies) and measures of scarring by periodic orbits (as the measure that we defined in [46] (Pilatowsky 2020)).

With respect to the above-mentioned measures, let us add that:

(i) Localization Measures: The Rényi-Wherl entropies are Rényi entropies associated with the Husimi function. Our measure \mathfrak{L} is related to the Rényi-Wherl entropy of order 2. In the newly added reference [51] (Wang 2020), it has been shown that for the very same model that we study, a version of the Rényi-Wherl entropy of order 2 is linearly related to that of order 1. We have added this comment to the text below equation (3).

(ii) Scarring Measures: It is possible to quantify the degree of scarring of the eigenstates. This was pointed out in our paper, but we have now further extended the discussion, while avoiding too many technical details, since Nature Communications targets a broad audience. We now added to the list of references our more technical work [46] (Pilatowsky 2020), recently uploaded to the arXiv, where we introduced a measure of scarring and used it for the Dicke model.

- “How general are these results (see below)?”

Reply. We have found comments suggesting that ubiquitous scarring may be quite general. In the newly added reference [21] (Muller 1994), that the Reviewer kindly shared with us, we found this quote for the hydrogen atom in a magnetic field

...about 90% of eigenstates may be unambig[u]ously related to fixed points and invariant manifolds of periodic orbits, indicating that scars are the rule rather than the exception.

and

We find scars up to the highest calculated eigenstate ($\hbar_{eff} \geq 0.03$). Even though not all Husimi distributions could be linked uniquely to the shortest 12 POs we practically found no wavefunction which is ergodically distributed over the irregular part of the phase space.

Also, in the new reference [23] (Revuelta 2020), they were able to build significant portions of the spectrum of a simple two-degree-of-freedom chaotic model using only periodic orbits of relatively short periods. This indicates that the structure of the eigenstates of this model may be described almost entirely by its classical periodic orbits, hinting that all eigenstates may be scarred.

A merit of our work is exactly to motivate the question of whether scarring in other models is also the rule and not the exception. Our work serves as a guideline for these future studies, it provides the tools for this search in other models.

- “While exemplary results presented are interesting no definite conclusions are formed, the manuscript brings observations and questions but few definite answers. In this respect it may stimulate discussion.”

Reply. Our work provides answers for the Dicke model and should indeed inspire similar discussions in other models. But our work has an additional purpose, which is to clarify concepts that have been misused in the recent literature on quantum scars. We significantly improved the Discussion Section and various parts of the manuscript to stress the differences and relationships between quantum scarring, quantum ergodicity and phase-space localization. We believe that in face of these explanations our conclusions will be better grasped.

- “However, the model studied is quite specific and studied by the same group in a number of papers”

Reply. The Dicke model is a simple interacting spin-boson model and has been theoretically studied in connection with various topics, from superradiance and quantum phase transitions to chaos and nonequilibrium quantum dynamics. As we wrote in the Introduction, it can also be studied experimentally. Furthermore, our phase-space techniques are applicable to any model with a phase space.

- “In the eighties scarring was studied not only for billiards or kicked top –incidentally for the latter the first reference on scars, prior to [13] is M. Kus, J. Zakrzewski and K. Zyczkowski, Phys. Rev. A43, 4244 (1991). More importantly, pioneering works on scarring in hydrogen atom in magnetic field, which really introduced periodic orbit theory to experiments should be cited (in particular works of late D. Wintgen, see e.g. D. Wintgen and A. Hönig Phys. Rev. Lett. 63, 1467 (1989) and later works e.g. K. Muller and D. Wintgen J. Phys. B 91994), works with the experimental group of late Welge). There is also an interesting Physics Reports from 1993 of Bohigas Tomsovic and Ullmo on a related topic, certainly worth a citation.”

Reply. We very much thank the Reviewer for these relevant references. We have incorporated all of them to our text.

- “Looking at Fig. 2 or supplementary material one hesitates whether the scarring is not enhanced by the relatively large effective $\hbar = 1/j$ in the model studied.”

Reply. This is a good point and we are glad to show that it is actually quite the opposite. As j increases, the orbits get better defined in the Husimi projections. We now added figures of the Husimi projections for different system sizes in the Supplementary Information. The patterns for $j = 30$ and $j = 100$, for example, are very similar, but the lines marking the periodic orbits become better delineated for $j = 100$.

- “Understanding the technical difficulties of the model one wonders whether a study in other models mentioned might not be more conclusive.”

Reply. We do believe our work will motivate extensions to other models, and our work provides the tools for that. Our method is applicable to any system with a phase space. With the analysis of various other models, one should eventually be able to build a general framework for the results presented in our work.

REPLY TO REVIEWER #2

We thank the Reviewer for carefully reading our manuscript and for thinking that our results are counterintuitive and potentially very interesting. The Reviewer raises important questions, which we address in detail below. They also motivated several improvements to the manuscript.

Point-by-point reply:

- “The authors just demonstrate the results without much explanation. For example, there is no explanation of why all the eigenstates encode the unstable periodic orbitals in the classical limit, which is the most striking result.”

Reply. We extended our explanations of the main concepts – quantum scarring, phase-space localization, and quantum ergodicity – which have connections and also differences. This is done in different parts of the text and also in the Discussion Section.

To provide a quantitative demonstration that all eigenstates encode the unstable periodic orbits, we need to use a measure of scarring. We do not give a direct measure of scarring (phase-space concentration around classical periodic orbits) in this article, because it is rather technical, and this Nature Communications paper targets a broad audience. However, we have now added more discussions about the subject and included reference [46] (Pilatowsky 2020), which we recently uploaded to the arXiv. In this more technical work, we introduced a direct measure of scarring. We elaborate more on this topic on the reply to the Reviewer’s point 3.

- “Furthermore, because of the phase space method that the authors use, it is difficult to assess whether the results are tied to this particular method or are intrinsic about the eigenstate in the Dicke Model.”

Reply. Our phase-space method is applicable to any quantum system that has a tractable phase-space (see the reply to the bullet below). Our work provides a guideline on how to apply the measures that we define to any of these systems.

We would also like to bring to the Reviewer’s attention that we have now found indications in the literature that our results should indeed be general. For example, in the newly added reference [23] (Revuelta 2020), they are able to reconstruct significant portions of the spectrum of a chaotic quantum system using only periodic orbits. This means that these eigenstates are entirely described by those periodic orbits and are therefore scarred. In another reference – [21] (Muller 1994) – also added to our list, the authors suggest that for the hydrogen atom in a magnetic field, at least

90% of eigenstates may be unambig[u]ously related to fixed points and invariant manifolds of periodic orbits, indicating that scars are the rule rather than the exception.

In this reference we also read these surprising statements,

We find scars up to the highest calculated eigenstate ($\hbar_{eff} \geq 0.03$). Even though not all Husimi distributions could be linked uniquely to the shortest 12 POs we practically found no wavefunction which is ergodically distributed over the irregular part of the phase space.

These claims indicate that ubiquitous scarring may be present in many other models. A great merit of our work is to provide the tools to visualize the phenomenon of scarring and verify these claims.

- “As the authors also admit, it is not obvious how to identify the phenomena observed here in other quantum systems like spin 1/2 models, where the phase space method is not applicable.”

Reply. We share the Reviewer’s concern, but stress that quantum scarring is intrinsically a phase-space effect, since it is defined as the concentration of quantum states around classical periodic orbits in the phase space. The recent discussions about many-body quantum scars are very exciting, but these studies are still in need of the phase-space analysis to be conclusive. The semiclassical analysis of many-body quantum systems is extremely difficult due to the proliferation of periodic orbits. However, works by Thomas Guhr [e.g. PRL 118, 164101 (2017)] and Klaus Richter [e.g. PRL 121, 124101 (2018)] give us good reasons to be optimistic. The preliminary studies about the semiclassical dynamics of the PXP model may also bear fruits [see arXiv: 2011.09486]. And, as we mentioned above, if the system has a tractable phase space, our methods are applicable.

- 1) “The measure L introduced in eq (3) to quantify the localization in phase space is not enough to claim that all eigenstates are scarred. None of the pure states occupy the whole phase space, including the random states. L of many eigenstates in the high energy shell is comparable with that of random states ~ 0.5 .”

Reply. We are glad the Reviewer made this comment, because it motivated us to improve our explanations in the text. Let us go by parts.

– First, \mathfrak{L} is not a measure of scarring, but of *phase-space localization*. The claim that all eigenstates are scarred is based on the Husimi projections. We show that in contrast to the random states, the Husimi projections of all eigenstates in the chaotic region show structures that look like periodic orbits. Compare the Husimi projections of the random states in Fig. 2 (r1)-(r4), which “do not show structures that resemble closed periodic orbits”, with Fig 2 (s1)-(s22) and the additional plots in the Supplementary Information. [See also our comments about how to quantify the degree of scarring of the eigenstates in our reply to the Reviewer’s point 3.]

– It is of course true that there is a relationship between phase-space localization and scarring. An eigenstate that is scarred by one periodic orbit of a family of periodic orbits is also very localized, but there are eigenstates that may be scarred by more than one periodic orbit and from different families. This is shown quantitatively in

the technical reference [46] (Pilatowsky 2020), but we now see the importance of qualitatively clarifying this to the Nature Communications’ readers as well, so discussions were added.

– The Reviewer is very much right that there are some eigenstates with $\mathfrak{L}_k \sim 1/2$, that is, with degrees of delocalization comparable to the random states, but look how interesting this gets! We now added panels (b) and (c) to Fig. 2. They show the distributions of \mathfrak{L} for 20,000 random states [Fig. 2 (b)] and for the eigenstates in the chaotic region [Fig. 2 (c)] for various different system sizes j .

→ In the case of random states, the values are concentrated around $1/2$ and the width decreases as j increases.

→ In contrast, for the chaotic eigenstates:

(i) the distribution is skewed and broader,

(ii) the tail at small values of \mathfrak{L}_k does not change as j increases, showing that the highly scarred states persist,

(iii) the portion of the states with large \mathfrak{L}_k (that is $\mathfrak{L}_k = 1/2$ or slightly larger) decreases, suggesting that for sufficiently large system sizes, none of the eigenstates would reach the level of delocalization of random states. The cases $\mathfrak{L}_k \sim 1/2$ that we now see, such as in Fig. 2 (s8), may be a finite-size effect. Although, even here, as we said above and in the main text, the pattern of periodic orbits is clearly visible, while this is not the case for the random states.

- 2) “The time-averaged L introduced below eq (3) is also not a good measure to claim all eigenstates are scarred. Quantum mechanics determines that the eigenstate does not have dynamics and does not explore the phase space, in contrast to a random state. This measure leads to that all eigenstates of all quantum systems are scarred, which is not meaningful.”

Reply. First, it is important to make it clear that $\overline{\mathfrak{L}}$ is not a measure of scarring, it is a measure of quantum ergodicity. $\overline{\mathfrak{L}}$ “quantify[ies] how much of the energy shell is visited on average by the evolved state.” This idea is analogous to the notion of ergodicity in the classical limit. The fact that $\overline{\mathfrak{L}}$ is below $1/2$ for all eigenstates does not tell us that they are scarred. As we wrote below Eq.(4), this means that “all stationary states in the chaotic region of the Dicke model are non-ergodic”.

Let us rephrase the Reviewer’s sentence,

“This measure leads to that all eigenstates of all quantum systems are scarred”

and use instead,

“Since all eigenstates of quantum systems cannot fill the whole *phase-space*, they are non-ergodic”.

That \mathfrak{L}_k has to be smaller than 1 for any pure state (including eigenstates) is due to interferences and to the fact that the Husimi functions always have zeros (nodes) [JPA 30, L677 (1997); JPA 34, 10123 (2001).]

We significantly improved the Discussion Section and various parts of the manuscript to stress the differences and relationships between quantum scarring, quantum ergodicity

and phase-space localization.

- 3) “The most interesting and surprising result I find is that all eigenstates encode the pattern of the unstable periodic orbital, different from a random state. However, the authors did not discuss the mechanism behind this or provide a quantitative measure to quantify the eigenstate’s closeness to the periodic orbital. Is there a quantitative measure to distinguish the pattern of phase space in Fig 2 (r4) and (s22), which have a similar L ?”

Reply. Just as the Reviewer, we also found this result interesting and surprising. The answer to the question is yes, there is a measure of scarring. We introduced it in our more technical paper [46] (Pilatowsky 2020). It is the measure \mathcal{P} in equation (18) of that paper.

Let us provide some explanations, which we also added to the manuscript. To measure the degree of scarring of all the eigenstates one has to know all the classical periodic orbits that generate the scars. Finding all orbits is extremely challenging. In the technical paper [46] (Pilatowsky 2020), we were able to find two families of periodic orbits and thus directly measure the degree of scarring generated by them. Interestingly, we found that there are eigenstates scarred by periodic orbits from both families. These are also the states with smaller values of \mathfrak{L}_k . We also note that:

- The orbits from the families identified in [46] (Pilatowsky 2020) are the ones visible in the eigenstates of Fig.1 of our paper.
 - We have not identified the periodic orbits in Fig. 2, but those circular patterns in the Husimi projections are clear evidence of periodic orbits. Their existence is clear from the shape of the Husimi projections and by knowing the generic direction of the classical Hamiltonian flow. The patterns display all the features of periodic orbits: they always cross the line $P = 0$ perpendicularly, they display symmetry along the P and Q axes, and they visibly form closed loops. There is no quantum effect other than scarring that would produce such patterns.
 - According to our measure \mathcal{P} of scarring, random states have $\mathcal{P} \sim 1$, while all eigenstates should have $\mathcal{P} > 1$. We say “should have”, because to say “have”, we would need *all* periodic orbits. For the eigenstates scarred by the periodic orbits that we found, then indeed we verified that $\mathcal{P} > 1$, as seen in Fig. 2 of the technical paper.
- 4) “Can the result found be a finite-size effect? How does the result change as ω_o increase?”

Reply. After reading our answer to the point 1 above, where we described the distributions for the phase-space localization measure for various system sizes, the Reviewer may already be convinced that our results are not finite-size effects. But to further reinforce this fact, let us address this question also in terms of scars. The orbit-like patterns visible in the Husimi function of the eigenstates are visible for all values of j numerically accessible and the lines get better defined as the system size increases.

We added Husimi distribution for different systems sizes to the Supplementary Information.

Note: We believe the Reviewer meant to say j , not ω_0 , but just in case, we clarify our choice of value for this parameter. We work in resonance ($\omega = \omega_0$), because this is where the chaotic regime of the model is the largest. Increasing ω_0 , and thus moving out of resonance, would reduce the energy span of the chaotic regime and more eigenstates would fall in the regular regime. In the limit of $\omega_0 \rightarrow \infty$ the system becomes integrable.

- 5) “In recent studies of quantum many-body scars related to the long-time oscillation found in the Rydberg atom experiments, a Hallmark of quantum scar is that order L eigenstates violate eigenstate thermalization hypothesis (ETH) while the level statistics of the whole spectrum remains Wigner-Dyson like. Do the scarred states identify here, namely all eigenstates, violate ETH? This can be checked by comparing the local observables measured from eigenstates and the Gibbs ensemble. What does the projected Husimi distribution of a Gibbs state look like? ETH violation can be used as a measure independent of the phase space method to strengthen the authors’ claim.”

Reply. From the way this question is asked, we believe the Reviewer is familiar with what we write below, but still, for completeness, we choose to go through this answer in detail. The short answer could go as follows.

In the analysis of the Peres lattice, that is, the analysis of the plot of the eigenstate expectation value (EEV) of a few-body observable $O_{\alpha\alpha} = \langle \alpha | O | \alpha \rangle$ vs the eigenvalues E_α ¹ that is commonly used in studies of ETH, we should expect larger fluctuations for the eigenstates that are highly scarred, but separating these states from the eigenstates that are not so strongly scarred is not straightforward. In addition, the comparison with thermodynamic averages has to be followed by scaling analysis and the range of system sizes numerically accessible is often very limited. We provide some illustrations below, but reiterate that studies of scarring can only be conclusive if followed by the analysis of phase space.

We added a sentence about ETH in the Discussion Section, because we recognize that this should be of interest to the community studying “quantum many-body scars”, but a complete discussion of this subject requires extended explanations and analysis, which we may present in a future publication focusing specifically on scarring vs ETH and thermalization.

Let us now move to the details.

Thermalization:

- (i) ETH has two aspects: diagonal and off-diagonal ETH. To simplify the discussion, we focus here only on the first, since this seems to be what the Reviewer has in mind. The diagonal ETH says that when the EEV of a few-body observable $O_{\alpha\alpha}$ obtained with an eigenstate $|\alpha\rangle$ does not vary much for eigenstates close in energy, then that

¹This kind of plot was discussed by Peres in the 1980’s, although the ETH community is usually not aware of this name.

observable should thermalize ². Quantum chaos in the sense of chaotic eigenstates ³ is the mechanism that guarantees the validity of ETH [see e.g. Physics Reports 626, 1 (2016)].

(ii) To talk about *thermalization*, we should, of course, add to (i) information about the initial state. Thermalization will happen for initial states with energies within that window of energy, where $O_{\alpha\alpha}$ does not vary much, that is, initial states that are large superpositions of those chaotic states [see e.g. Physics Reports 626, 1 (2016)].

(iii) There is another scenario where thermalization will also happen: When despite having an integrable model, where the eigenstates are not chaotic, the initial state itself is chaotic in the energy eigenbasis, that is, it is an uncorrelated superposition of very many eigenstates [see e.g. PRL 108, 110601 (2012)].

Peres lattice:

To verify (i), it is not enough to simply look at the Peres lattice, one needs a quantitative analysis as well. In Fig. 2 of [46] (Pilatowsky 2020), the Reviewer can see the entire Peres lattice of an observable of the Dicke model: the regular region (low energies) and the chaotic region (large energies) are clearly distinguishable. In the figure below, we show that Peres lattice for three different system sizes j (the observable, $O = n_e$, is the number of excited atoms). It is hard to say whether the fluctuations in the chaotic region are decaying or not as j increases.

Following PRE 82, 031130 (2010), let us then look at the deviation of the EEVs with

²(provided the off-diagonal ETH is also satisfied)

³(that is, states that are close to random states, although never equal to them, because correlations are always present in realistic systems)

respect to the microcanonical result O_{mic} [Eq.(12) in PRE 82, 031130 (2010)],

$$\Delta^{\text{mic}}O = \frac{\sum_{\alpha} |O_{\alpha\alpha} - O_{\text{mic}}|}{\sum_{\alpha} |O_{\alpha\alpha}|}$$

The figure below, obtained within the chaotic region, does suggest the validity of ETH, as expected for chaotic systems. We see that $\Delta^{\text{mic}}O$ does decrease as j increases.

However, the best quantitative analysis, specially in a scenario where highly scarred states may be present, is to deal with the normalized extremal fluctuation [Eq.(13) in PRE 82, 031130 (2010)],

$$\Delta_e^{\text{mic}}O = \left| \frac{\max O - \min O}{O_{\text{mic}}} \right|,$$

where the maximum $\max O$ and minimum $\min O$ values of $O_{\alpha\alpha}$ are extracted from the same window of energy used to obtain the microcanonical value. Now, for this quantity, things become fuzzier, most likely due to the presence of the highly scarred states, as seen in the figure below.

In short, all of our eigenstates are scarred, but they have different degrees of scarring. The highly scarred eigenstates, which are also the most localized ones, should indeed lead to large fluctuations in the Peres lattice. The eigenstates that are more delocalized and thus closer to random states (although they are still scarred, usually by periodic orbits of different families) should lead to smaller fluctuations in the Peres lattice.

Initial State:

To verify whether thermalization indeed takes place at long times, then we need to also take (ii) into account, that is, we need to investigate the structure of the initial state with respect to the energy eigenbasis. Remember what we said in (iii), that depending on the initial state, thermalization may happen even in an integrable model. The opposite is also true, depending on the initial state, thermalization may not happen even in a chaotic system.

In our case, the initial states are coherent states.

- If the coherent state is centered at an unstable periodic orbit of short period, then it is highly scarred. This initial state is a superposition of not too many eigenstates, which in turn are highly scarred by that periodic orbit. For this kind of initial state, the local density of states (that is, the energy distribution of the initial state) shows the typical comb-like structure discussed in the early works by Heller, the survival probability (SP) leads to large oscillations, and the infinite-time average of the SP reaches large values, in other words, thermalization will not happen.
- On the other hand, if the coherent state is far from the periodic orbits of short period, then it becomes a large superposition of eigenstates that are scarred by different periodic orbits and by periodic orbits belonging to different families. The comb-like structure and the oscillations of SP are no longer seen, and the infinite-time average of the SP reaches values close to the IPR of Gibbs states, in other words, thermalization will happen.
- This discussion (although not mentioning thermalization explicitly) can be found in

the Sec. IV of [46] (Pilatowsky 2020). It is, of course, straightforward to extend this sort of analysis to few-body observables.

Husimi: Answering the question “What does the projected Husimi distribution of a Gibbs state look like?”, a Gibbs state at infinite temperature is equivalent to an average over random states, so its Husimi projection should be similar to what we see in Fig. 3 (h3). Indeed, we show below the Husimi projection and $\mathcal{L}(\epsilon, \hat{\rho}_G)$ for a Gibbs state, where the weights are all equal

6) “Is there any experimentally measurable consequence of these findings?”

Reply. This is a good question, which we had also asked ourselves. We should certainly expect measurable effects for the dynamics of strongly scarred coherent states, like the one in Fig. 3 (a1), for which the survival probability exhibits revivals [Fig. 3 (a2)]. But the Reviewer probably wants to know if one could experimentally detect the “ubiquitous scarring” that we discuss in this paper. This would require both finding a model where scarring is ubiquitous (and as we explained above, this may be very general) and an experiment that has access to the wave functions. What is known in the literature is that microwave cavities allow for the experimental study of billiard wave functions [see e.g. *Quantum Chaos: An Introduction* by Hans-Jürgen Stöckmann]. That is how far we can go with this answer at the moment.

Reviewers' Comments:

Reviewer #1:

Remarks to the Author:

I find the replies of authors quite satisfactory. The manuscript is interesting, provides strong numerical evidence for scarring discussed and may be quite useful for the broad community interested in classical-quantum correspondence. I recommend publication.

Reviewer #2:

Remarks to the Author:

The authors have answered both referees' questions in great detail. The quality of the revised manuscript is improved. In the new version, the authors have clarified the definition of the scar and phase space localization. They also included the distribution of L of eigenstates and random states, as well as the time average of L of random states and coherent states. Even though the authors do not provide an answer to the found excessive scarring, I do find that the finding that all eigenstates are related to the periodic orbital is interesting. It seems to indicate the structure of eigenstates beyond ETH since the Husimi function probes nonlocal properties of the eigenstates. The result presented in this paper may lead to some further interesting study. Therefore I'm positive toward its publication in Nature Communication.

Reply to Reviewers:

We thank both reviewers for recommending publication of our work in its present form. None of the two made any new suggestions for changes or improvements.